# Influence of Occupational Stress and Coping Style on Periodontitis among Japanese Workers: A Cross-Sectional Study

**DOI:** 10.3390/ijerph16193540

**Published:** 2019-09-22

**Authors:** Md Monirul Islam, Daisuke Ekuni, Toshiki Yoneda, Aya Yokoi, Manabu Morita

**Affiliations:** Department of Preventive Dentistry, Okayama University Graduate School of Medicine, Dentistry and Pharmaceutical Sciences, Okayama 700-8558, Japan; p3a99o50@s.okayama-u.ac.jp (M.M.I.); dekuni7@md.okayama-u.ac.jp (D.E.); de17057@s.okadai.jp (T.Y.); mmorita@md.okayama-u.ac.jp (M.M.)

**Keywords:** occupational stress, coping, periodontitis, Japanese workers

## Abstract

The aim of this cross-sectional study was to evaluate the association between the influence of occupational stress and coping style on periodontitis among Japanese workers. The study sample included 738 workers (age range: 19–65 years) at a manufacturing company in Kagawa Prefecture, Japan. To analyze occupational stress and coping style, all participants answered a self-report questionnaire composed of items on their work environment and oral health behavior. Oral examinations were performed by calibrated dentists. Among all workers, 492 (66.7%) workers were diagnosed with periodontitis, and 50 (6.8%) were diagnosed with a high stress-low coping condition. Significant differences (*p* < 0.05) were observed between the periodontitis and non-periodontitis groups in terms of age, gender, body mass index, smoking status, daily alcohol drinking, monthly overtime work, worker type, and stress-coping style. Logistic regression analysis showed that a high stress–low coping condition was associated with an increased risk of periodontitis (odds ratio: 2.79, 95% confidence interval: 1.05–7.43, *p* = 0.039). These findings suggest that a high stress-low coping condition is associated with periodontitis among the 19–65 years of age group of Japanese workers.

## 1. Introduction

In recent years, occupational stress has become an increasingly serious problem around the world [1]. Remarkable changes in working duration, job engagement, and type of working environment have led to increased levels of occupational stress in workers [2]. The World Health Organization considers occupational stress among workers a global epidemic and is actively seeking to ascertain its severity [3]. Occupational stress is becoming a markedly serious problem among Japanese workers. According to previous studies, more than 60% of Japanese workers have reported experiencing work-related stress, and the number of workers with mental health problems has been rapidly increasing [4,5].

Occupational stress has adverse effects on employment (e.g., absenteeism, lateness, job dissatisfaction, and job turnover) [6,7,8]. Moreover, it contributes to poor physical and mental health [9]. For instance, research has shown that occupational stress is associated with certain oral health problems, including caries [10], periodontal disease [11], temporomandibular disorder, and halitosis [12]. Stress has long been regarded as an important predisposing factor for periodontal disease among workers, and an association has been reported between periodontal disease and work-related stress and dissatisfaction [13]. Coping is the response of the individual to control, minimize, or avoid the adverse and unpleasant effects of stress. According to Lazarus et al., an individual’s psychological and physical well-being depends on coping strategies more than the frequency and intensity of stress [14,15]. Concurrently, the relationship between stress and individual coping styles has been shown to be associated with periodontal disease. A 24-month prospective study involving chronic periodontitis patients found that patients with an active coping style showed a lesser degree of disease advancement than those with a passive coping style [16]. Furthermore, the effects of stress on periodontal disease can be restrained by adequate coping behaviors [17]. Another study found that coping style had a protective effect against tooth loss [18]. However, to our knowledge, no studies have explained the influence of coping style against stress, especially workplace-induced occupational stress, on periodontal disease.

Therefore, the aim of the present cross-sectional study was to investigate the influence of occupational stress and coping style on periodontitis among Japanese workers, and, thereby, test the hypothesis that the balance between occupational stress and coping style is related to periodontal disease.

## 2. Materials and Methods

### 2.1. Study Design and Participants

Anonymous data for this cross-sectional study were obtained from the workers of a Japanese crane manufacturing company located in Kagawa Prefecture, Japan, for the years 2016–2018 (in February each year). All workers received a routine general health checkup, including a voluntary dental checkup. We distributed the questionnaires to the participants prior to health checkups and collected them during the annual health checkups. A total of 1476 workers participated in the general health examinations, among whom, those who received a voluntary dental checkup and completed a questionnaire (n = 855) were enrolled in the study. After excluding those with incomplete questionnaires, we analyzed the data of 738 (86.3%) workers. All participants provided informed consent at the time of their interview.

### 2.2. Job Category

We followed the job category criteria provided by the Ministry of Health, Labor, and Welfare of Japan. According to these criteria, all participants were classified as office workers (e.g., computer operators, clerks, secretaries) or skilled workers (e.g., factory workers, construction workers).

### 2.3. Assessment of Stress and Coping Style

In this study, we used the “Co–Labo57 +” questionnaire [19], which is composed of six parts (parts A–F). Questions from parts A–D adopt the Brief Job Stress Questionnaire (BJSQ) to measure occupational stress. The reliability and validity of the BJSQ was confirmed for Japanese workers [12,20,21]. The BJSQ is composed of 57 items used to assess job stressors (Part A, 17 items: e.g., psychological job demands, job control), stress response (Part B, 29 items: e.g., psychological and physical stress reactions), and buffering factors (Parts C and D, 11 items: e.g., social support at work). The BJSQ program manual suggests criteria for categorizing stress levels [22]. High-stress was determined as having the highest level of a stress reaction [criterion (i)] or a moderate level of a stress reaction, along with having the highest job stressors (or lowest social support in the workplace) [criterion (ii)]. In this study, to calculate the stress reaction and job stressor scores, we summed the item scores from a four-point Likert scale (from 1 = low stress to 4 = high stress). The scores ranged from 29 to 116 for stress reactions (Part A) and from 26 to 104 for job stressors (Part B). The cutoff points were 77 for the stress reaction score for criterion (i), 63 for the stress reaction score, and 76 for the job stressor score for criterion (ii) [23]. If the score for criterion (i) or criterion (ii) was higher than the cutoff point, the participant was classified as high stress.

Coping style was assessed using the remaining two parts (E and F). The participant was considered to have a high coping style if the summed score was ≥ 20 for part E or ≥ 68 for part F.

### 2.4. General Health and Lifestyles Assessment

All participants underwent a mandatory general health examination. Body weight and height were measured to calculate body mass index (BMI), which was categorized as <25 or ≥25 kg/m^2^ [24]. In addition, the participants provided answers regarding their age, gender, and other lifestyle-related factors, including job type (office or skilled worker), smoking status (current, never, or former), daily alcohol drinking (yes or no), daily sleep duration (≥6 or <6 h) [25], and amount of overtime work (≥40 or <40 h/month) [26]. The amount of overtime work was assessed by calculating total hours worked minus the standard 8 working hours per day on weekdays, plus the number of hours worked on holidays during the entire month.

Oral health behavior was assessed by asking about the use of dental floss (yes or no) and whether the participant had a regular dental checkup in the past year (yes or no) [27].

### 2.5. Oral Examinations

The participants’ oral conditions (e.g., number of healthy, missing, and decayed teeth, plaque and calculus level, gingival and periodontal health, malocclusion, temporomandibular joint findings) were evaluated by calibrated dentists. Using an objective method [28], the participants were then classified into either a periodontitis or a non-periodontitis group. Briefly, no inflammation in the gingiva or redness and/or swelling of the interdental papilla without gingival recession was classified as non-periodontitis, and any redness and/or swelling in the gingiva with gingival recession and/or tooth mobility was classified as periodontitis. The intra and inter-examiner reliabilities was evaluated by kappa statistics of more than 0.8.

### 2.6. Ethical Considerations

The study protocol was approved by the Okayama University Graduate School of Medicine, Dentistry, and Pharmaceutical Sciences and Okayama University Hospital Ethics Committee (No. 1905-016).

### 2.7. Statistical Analysis

The normality of the data was investigated using a histogram, quantile–quantile plot, and the Shapiro-Wilk test [29]. The sample size was estimated using G * Power (ver. 3.1.9.2, Universität Kiel, Kiel, Germany), and the minimum sample sizes were calculated for analysis using a chi-squared test [30]. Considering an effect size of 0.3, alpha of 0.05, and power (1 − β) of 0.80, the minimum sample size required was 88 [31]. Since the company had more than an adequate number of workers to obtain reasonable results, we enrolled more than 88 workers in our study.

For the descriptive analysis, means and standard deviations were calculated for continuous variables, whereas categorical variables were presented as percentiles. *p* values were calculated for the continuous and categorical variables using the Mann-Whitney *U* test and chi-squared test, respectively. The results from the logistic regression analysis were presented as odds ratios (ORs) and 95% confidence intervals (CIs). *p* values < 0.05 were considered statistically significant. All analyses were performed using the SPSS statistical package (v. 25.0; SPSS Inc., Chicago, IL, USA).

## 3. Results

Figure 1 shows a flowchart of the enrollment procedure. Following the scoring criteria of the Co-Labo57+ questionnaire, we identified a total of 88 (11.9%) workers as high-stress and 438 (59.3%) as having a high coping style. The characteristics of the workers are shown in Table 1. Among the 738 workers, 646 were men (87.5%) and 92 were women (12.5%). The mean age was 40.7 ± 10.5 years. The prevalence of periodontitis was 66.7% (n = 492). Among the workers, 88.1% (n = 650) were classified to the “low stress,” 5.1% (n = 38) to the “high stress–high coping,” and 6.8% (n = 50) to the “high stress–low coping” group.

Table 2 shows a comparison between the periodontitis and non-periodontitis groups. The mean age in the periodontitis group was significantly higher than that in the non-periodontitis group (*p* < 0.001). Significant differences were also seen between the two groups in gender, BMI, smoking status, daily alcohol drinking, monthly overtime work, worker type, and stress-coping style (*p* < 0.05).

As shown in Table 3, the logistic regression analysis revealed that periodontitis was significantly associated with age, the male gender, BMI ≥ 25 kg/m^2^, and smoking status (current smoker) (*p* < 0.001). Moreover, periodontitis was significantly associated with the “high stress-low coping” style (*p* = 0.039).

## 4. Discussion

In this cross-sectional study, we focused on the influence of occupational stress and coping style on periodontitis. To the best of our knowledge, this is the first study to investigate whether the balance between occupational stress and coping style is associated with the prevalence of periodontitis among Japanese workers. Our results showed that having a low coping style against high occupational stress was significantly associated with periodontitis (adjusted ORs: 2.79, 95% CI: 1.05–7.43, *p* = 0.039).

The workers with the high stress–high coping style were not found to be at a higher risk for periodontitis in this study, which suggests an association between periodontitis and an effective stress-coping style. This condition can be explained by several possible mechanisms. For example, an inappropriate coping attitude (e.g., increased tobacco smoking, alcohol drinking) and the adoption of unhealthy behaviors (e.g., poor oral hygiene, insufficient nutritional intake) can lead to drastic changes in oral health [32,33,34]. Concurrently, an inadequate stress-coping style may cause mental alterations and induce immune suppression, which may aggravate chronic inflammatory diseases such as periodontitis [35].

Our results found a higher prevalence of current smokers with periodontitis than in the non-periodontitis group. Previous studies have reported smoking as a coping mechanism against stress among workers [36,37]. On the other hand, an association between smoking and the risk of periodontitis has been established [18,38,39]. Therefore, attempts should be made to prevent smoking among workers.

In the present study, no significant difference in overtime work was found between the periodontitis and non-periodontitis groups. A previous study among Korean workers reported that overtime work was associated with the prevalence of periodontitis [40]. The discrepancy in the findings from the previous study and the present study may depend on race (Korean vs. Japanese) and the percentage of overtime workers (60.3% vs. 25.3%). In addition, no significant difference in job category was found between the periodontitis and non-periodontitis groups, which is in line with previous studies finding no significant association between the job category (skilled and office workers) and periodontitis [41,42].

Additionally, in line with a previous study [43], the overweight factor was found to be a risk factor for periodontitis among workers. Other previous studies have reported finding a positive relation between coping style and BMI [44,45]. Hence, to help find balance between stress and coping, workers should be encouraged to limit their working hours and engage in a healthy lifestyle while maintaining a normal body weight.

The prevalence of periodontitis in this study was 66.7%, which differed from other studies in Japan. A five-year cohort study that defined periodontitis as having one or more sextants with a Community Periodontal Index score > 2 (pockets ≥ 4 mm deep) reported that 55.4% of Japanese workers had periodontitis [41]. Another cross-sectional study that measured periodontitis by probing pocket depth and clinical attachment loss (CAL) at mesio-buccal and mid-buccal sites for all of the teeth in two randomly selected quadrants indicated that only 13% of Japanese workers had periodontitis, which was defined as having at least one tooth with a CAL of ≥7 mm [46]. Possible causes for this discrepancy might be the type of oral examination or the study design. On the other hand, in the present study, 11.9% of workers reported having high occupational stress. This result was in line with a previous Japanese study using the same questionnaire [23].

The present study showed that the risk of periodontitis is influenced by the balance between occupational stress and coping style. Therefore, a comprehensive approach should be taken to minimize the occupational stress and improve the coping ability of the workers. Our results also suggest that, during the management of periodontitis in a clinical setting, the coping style of workers against occupational stress should be taken into consideration.

This study had some limitations. First, all workers were enrolled from a Japanese crane manufacturing company. Therefore, the influence of stress and coping style on periodontitis might differ from other types of workers. Second, no other possible confounders, such as working schedule [40], education level [3], income level [47], and family situation [48], were examined in this study. Third, periodontitis was diagnosed using an objective method due to time constraints, but no probing or X-ray findings were used. Hence, there is a possibility of presenting a large group of mild periodontitis with little clinical impact. Lastly, a causal association could not be shown because this study was cross-sectional.

## 5. Conclusions

The findings of the present study suggest that a high stress–low coping style is associated with an increased risk of periodontitis among the 19–65 years of age group of Japanese workers. Therefore, during the management of periodontitis, the balance between occupational stress and coping style should also be considered.

## Figures and Tables

**Figure 1 ijerph-16-03540-f001:**
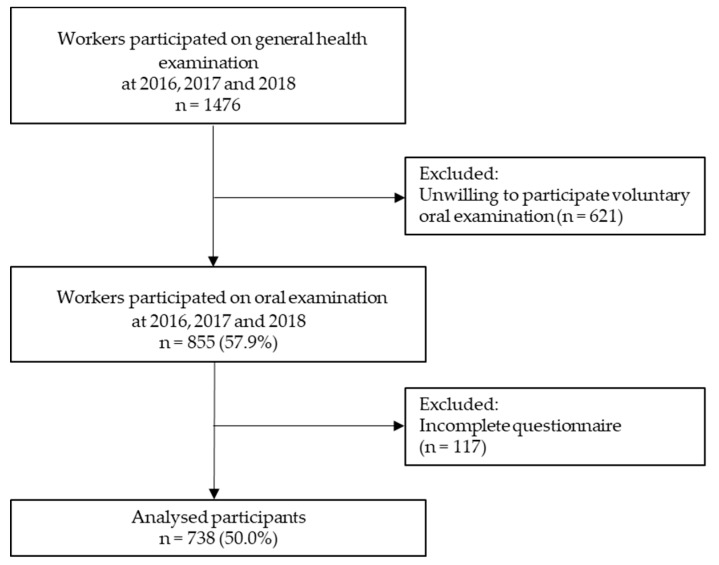
Flowchart of the enrollment procedure.

**Table 1 ijerph-16-03540-t001:** Characteristics of all workers.

Parameters (n = 738)	n (%) Mean ± SD
Age (y)	40.7 ± 10.5
Gender	
Male	646 (87.5)
Female	92 (12.5)
Daily flossing (Yes)	95 (12.9)
Regular dental checkup (Yes)	107 (14.5)
Periodontitis (Yes)	492 (66.7)
BMI (kg/m^2^)	
<25	545 (73.8)
≥25	193 (26.2)
Hypertension (Yes)	47 (6.4)
Daily sleeping duration (≥6 h)	631 (85.5)
Current smoker (Yes)	206 (27.9)
Daily alcohol drinking (Yes)	128 (17.3)
Monthly overtime work (≥40 h)	187 (25.3)
Worker type	
Skilled worker	334 (45.3)
Office worker	404 (54.7)
Low stress	650 (88.1)
High stress-High coping	38 (5.1)
High stress-Low coping	50 (6.8)

BMI, body mass index. SD, standard deviation.

**Table 2 ijerph-16-03540-t002:** Comparisons of the between periodontitis and non-periodontitis groups.

Parameters (n = 738)	Periodontitis (n = 492)	Non-Periodontitis (n = 246)	*p* Value
	n (%) Mean ± SD	n (%) Mean ± SD	
Age (y)	43 ± 10.1	34.9 ± 8.9	<0.001 ^1^
Gender (Male)	462 (93.9)	184 (74.8)	<0.001 ^2^
Daily flossing (Yes)	64 (13.0)	31 (12.6)	0.876 ^2^
Regular dental checkup (Yes)	70 (14.2)	37 (15.0)	0.767 ^2^
BMI (kg/m^2^)			
<25	336 (68.3)	209 (85.0)	<0.001 ^2^
≥25	156 (31.7)	37 (15.0)	
Sleeping duration (daily)			
≥6 h	418 (85.0)	213 (86.6)	0.554 ^2^
<6 h	74 (15.0)	33 (13.4)	
Current smoker (Yes)	161 (32.7)	45 (18.3)	<0.001 ^2^
Daily alcohol drinking (Yes)	104 (21.1)	26 (10.6)	<0.001 ^2^
Monthly overtime work			
(≥40 h)	145 (29.5)	42 (17.1)	<0.001 ^2^
Worker type			
Skilled worker	245 (49.8)	89 (36.2)	<0.001 ^2^
Office worker	247 (50.2)	157 (63.8)	
Low stress	431 (87.6)	219 (89.0)	<0.001 ^2^
High stress-High coping	17 (3.5)	21 (8.5)	
High stress-Low coping	44 (8.9)	6 (2.5)	

BMI, body mass index. SD, standard deviation. ^1^ Mann–Whitney *U* test, and ^2^ chi-squared test.

**Table 3 ijerph-16-03540-t003:** Adjusted odds ratios (OR) and 95% confidence intervals (95% CI) for periodontitis.

Dependent Variable	Independent Variable		OR	95% CI	*p*-Value ^1^
Periodontitis	Age		1.11	1.09–1.14	<0.001
	Gender	Female	Ref		
		Male	5.11	2.81–9.30	<0.001
	Daily flossing	Yes	Ref		
		No	0.99	0.57–1.76	0.990
	Regular dental checkup	Yes	Ref		
		No	1.17	0.69–1.99	0.562
	BMI (kg/m^2^)	<25	Ref		
		≥25	2.23	1.42–3.51	< 0.001
	Sleeping duration (daily)	≥6 h	Ref		
		<6 h	0.98	0.58–1.67	0.938
	Current smoker	No	Ref		
		Yes	2.08	1.35–3.22	< 0.001
	Daily alcohol drinking	No	Ref		
		Yes	1.24	0.73–2.11	0.424
	Monthly overtime work	<40 h	Ref		
		≥40 h	1.07	0.68–1.71	0.765
	Worker type	Office worker	Ref		
		Skilled worker	1.31	0.89–1.95	0.175
	Low stress		Ref		
	High stress-High coping		0.30	0.14–0.66	0.003
	High stress-Low coping		2.79	1.05–7.43	0.039

BMI, body mass index. CI, confidence interval. OR, odds ratio. ^1^ Multiple logistic regression model adjusted for age, gender, daily flossing, regular dental checkups, BMI, sleeping duration (daily), smoking status (current smoker), daily alcohol drinking, monthly overtime work, worker type, and stress-coping style.

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
