# Peer review of "Influence of Occupational Stress and Coping Style on Periodontitis among Japanese Workers: A Cross-Sectional Study"

_ijerph, 2019, doi:10.3390/ijerph16193540_

Round 1

Reviewer 1 Report

This article evaluated the association between the influence of occupational stress and coping style on periodontitis among Japanese workers using a self-reported questionnaire at a manufacturing company in Kagawa Prefecture in Japan. Authors concluded that a high stress-low coping condition is associated with periodontitis among Japanese workers. From my point of view, this article is well presented and clearly described and the proposed methodology is proper and scientifically correct. Nevertheless, I have to put forward some question and suggestion as follows:

The significance of the study is not clear. The first study estimated the influence of coping style on periodontitis among Japanese workers was not enough to be the main significance of this study. The author should add what is advantage of this study, what reference values for clinic and public health, and what basis this results can provide for the prevention and control of periodontitis disease in the future. The research time is vague. The author mentioned the year covering 2016-2018, but doesn’t specify the specific date and reader can’t know whether it is a complete three-year period. Therefore, the specific research time should be added in this study. The manuscript explained the participants were divided into periodontitis and non-periodontitis groups according to an objective method. Can you explain why this method is used and why the study don’t use the professional diagnostic criteria and diagnostic codes to confirm whether participants having periodontitis disease. It is not a good decision to investigate the data normality using a histogram and Q-Q plot. It is suggested that authors use more objective statistical test to verify whether the data obey normal distribution. The format of all the tables in the article is incorrect. It is suggested that the three-line form of statistical table should be strictly followed. Why are there two table headers in Table 3? Did you have two logistic regressions? The authors need to explain this. In addition, the factors in the Table 1-3 should be classified, such as baseline information, copying styles, stress-copy style, etc. The factors in the table and the descriptions in the text need to correspond, such as daily alcohol drinking in the table and daily alcohol consumption in the text. It is necessary to complete the conclusion with more than one sentence and the format of references need further modification.

Reviewer 2 Report

The study evalueted the association between the influence of occupational stress and coping style on periodontitis among Japanese workers. It’s an importante research to better know how the life stily can to influence the oral health. I have some observations:
1) Display the calibration values beteweenn examibers.  

2) Considering the categories studied (worker type, low stress, periodontitis, BMI, etc), explain if the sample size is suficiente to perform the statistical analysis.

Reviewer 3 Report

The authors performed  a cross-sectional study to evaluate the association between occupational stress and coping styles on periodontitis among Japanese workers. The study sample included 738 workers. Significant differences were observed between the periodontitis and non-periodontitis groups for age, gender, body mass index, smoking status, daily alcohol consumption, monthly overtime work, worker type, and stress-coping style. Logistic regression analysis showed that a high stress–low coping condition was associated with an increased risk of periodontitis (odds ratio: 2.79; 95% confidence interval: 1.05–7.43; P = 0.039).

Comments:

Although the manuscript is generally written well, the last sentence of the Abstract, which states that a high stress-low coping condition is associated with periodontitis in Japanese workers should better be removed, since the sample was from only one factory located in one prefecture of the country. Thus, the results should not be generalized for the whole of Japan.

The authors used a periodontitis definition that did not adhere to internationally accepted guidelines (i.e. CDC-AAP), resulting in a rather high periodontitis prevalence. The case definition thus seems to include a large fraction of mild periodontitis forms with little clinical impact. This fact should be added to the "limitations" listed in the Discussion section.

Round 2

Reviewer 1 Report

According to the previous comments and suggestions, the manuscript is well modified and clearly described. Nevertheless, there are some details that the authors should be modified.

1.     In the abstract and conclusion, the author should not write “in this group of Japanese workers”. It should specify which age group it is. It is suggested that the author should change it into “in the 19-65 years group of Japanese workers”.

2.     The tense of the whole article is inconsistent. For example, “In this study, we used the “Co-Labo57+” questionnaire, which is composed of six parts (parts A–F). Questions from parts A–D adopt the Brief Job Stress Questionnaire (BJSQ) to measure occupational stress. The reliability and validity of the BJSQ have been confirmed for Japanese workers”, the author is advised to confirm and revise it.

3.     In line 79-81, the form of a sentence should be consistent. It can be changed to “(Part A; 17 items: e.g., psychological job demands, job control), (Part B; 29 items: e.g., psychological and physical stress reactions), and (Parts C and D; 11 items: e.g., buffering factors, such as social support at work”.

4.     In line 136-137, the data information in the sentence “we identified a total of 88 (11.9%) workers as high-stress and 438 (59.3%) as 136 having a high coping style” were not shown in the table, why?

5.     The format of the references is inconsistent. Reference 28 has volumes, Reference 31 has two years and Reference 36 has no page information. It is suggested that the author confirm and revise them.

6.     If circumstances permit, it is suggested that the author add a questionnaire for workers in the supplemental materials.
